# Research on Land Use Changes and Ecological Risk Assessment in Yongjiang River Basin in Zhejiang Province, China

**Peng Tian** [1,2], **Jialin Li** [1,2,*], **Hongbo Gong** [3,*], **Ruiliang Pu** [4], **Luodan Cao** [2], **Shuyao Shao** [2], **Zuoqi Shi** [2], **Xiuli Feng** [2], **Lijia Wang** [2] and **Riuqing Liu** [2]

1 Donghai Institute, Ningbo University, Ningbo 315211, China; tppyang@163.com
2 Department of Geography & Spatial Information Techniques, Ningbo University, Ningbo 315211, China; aidushude@163.com (L.C.); vickyssy@163.com (S.S.); shizuoqi123@163.com (Z.S.); Fengxiuli@nbu.edu.cn (X.F.); ljwang2013@163.com (L.W.); liuruiqingwh@163.com (R.L.)
3 School of Law, Ningbo University, Ningbo 315211, China
4 School of Geosciences, University of South Florida, Tampa, FL 33620-5250, USA; rpu@usf.edu
* Correspondence: nbnj2001@163.com (J.L.); ghb19@163.com (H.G.);
 Tel.: +86-574-87609531 (J.L.); +86-574-86683038 (H.G.)

**Abstract:** Studying land use changes and ecological risk assessment in Yongjiang River Basin in Zhejiang Province, China, provides theoretical references for optimal configuration of land resources and maintaining stability of ecosystems. Given impacts of land use changes on landscape patterns in the Yongjiang River Basin, ecological risk assessment indexes were constructed and used to analyze temporal and spatial variation characteristics of ecological risk within different periods. Results show that (1) the construction land area was increased quickly, while the cultivated area decreased sharply. A prominent characteristic of land use changes was manifested by transforming cultivated area and forestland into construction land. The utilized degree of the land increased continuously. Spatially, the land utilized degree in northern regions was higher than that in southern regions and the degree in eastern regions was higher than that in western regions. (2) The ecological risk in the Yongjiang River Basin was intensified and the area of high ecological risk was expanded by 893.96 km$^2$. Regions with low and relatively low ecological risks concentrated in western and southern regions of the Basin, whereas regions with high ecological risks were mainly in northern and eastern regions. Landscapes in cities and towns at a high economic development level are highly sensitive to human activities. (3) Transformation of ecological risk is complicated. Land area with the ecological risk changing from a low level to a high level was 4.15 times that with the ecological risk changing from a high level to a low level. There were 15 transformation directions among different ecological risk regions.

**Keywords:** land use changes; ecological risk; Yongjiang River Basin

---

## 1. Introduction

Since the beginning of the 21st century, population and environmental resources pressures have been intensified under the background of high-speed consumption of resources [1]. Sustainable development is particularly important for various countries, especially the sustainability of natural environments [2]. It is not only a basis for the sustainable development of a national economy, but also a strategic resource for current and future human survival and sustainable social development [2,3]. Sustainable development research has great significance for the safety and stability of human society. A risk assessment focusing on future damage is applicable to assessing sustainability, finds sources of risk through analysis, cuts off factors that threaten the sustainable development of a region,

and maintains regional security and stability [3]. Therefore, an ecological risk assessment may analyze a relationship between regional risk sources and risk receptors through the ecological security framework of ecosystem health protection and ecological risk control, and help better understand a set of exposure factors of regional ecological environment, and thus effectively improve regional ecological sustainable development [4].

The ecological risk assessment is of a method to evaluate possible adverse consequences of an ecosystem and relevant components upon external disturbances [5,6]. The ecological risk assessment started from 1980s; the term of "ecological risk assessment" was firstly used in the USA Environmental Protection Agency (EPA). In the beginning, it was only used in the field of human health evaluation [7] and it was applied to the protection of human health by USA EPA, such as carcinogenic risk evaluation, teratogenicity risk assessment and extremely small-scaled evaluation objects [8,9]. In 1992, the USA EPA issued the ecological risk assessment framework and defined it as the possibility that evaluation of negative ecological effect may occur or is taking place. Such possibility is the consequence that the receptor is exposed to single or multiple stress factors [10]. In 1998, this framework was expanded and modified, forming the basic guidelines of current risk assessment [11]. After the 1990s, in the context of increasingly prominent environmental issues, attentions to risk assessment were shifted from human health evaluation to ecological risk evaluation, while the risk pressure factor was transformed from a single chemical factor to multiple chemical factors and events that may cause ecological risks [12,13]. Moreover, risk receptors also were expanded from the human body to populations, communities, ecosystems, and landscape levels in river basins [14–16]. For examples, Skaare et al. [5] conducted an ecological risk assessment of organochlorine pesticides in the Arctic and their results indicated that pesticides posed a serious threat to the population status, health, and safety of polar bears. Kienast et al. [17] used climate change as a source of risk to analyze effects of the climate change on vegetation and species richness [17]. In addition, they used animal foraging behavior as a risk source to analyze the potential long-term effects of animal outings on mountain forest resources [18]. Their study also demonstrated the diversification of risk sources. Since then, from the late 1990s to the beginning of the 21st century, the field of ecological risk assessment has begun to expand, entering a regional ecological risk assessment stage (at a river basin or larger scale). For this case, Adam et al. [19] conducted an ecological risk study on the Clinch River Basin in Tennessee, USA, which made it possible to apply the ecological risk assessment to large-scale watersheds. Shea et al. [20] integrated a number of examples to examine the relationship between stressor effects, exposure potential, and ecosystem risk, and detailed the process of ecological risk assessment. Munns et al. [21] incorporated ecosystem services into the framework of ecological risk assessment, and took into account the ecosystem's ability to serve humans. Harris et al. [22] assessed the risk levels of current watershed ecosystem services (human health, water quality, leisure, and fisheries) and reduced the threat to human health from watershed pollution. All of these studies suggested that the ultimate goal of ecological risk assessment would be to serve the society and the people. Only by linking ecological risks with human well-being can we maximize the value of ecological risk assessment.

The studies reviewed above also show that the risk sources and risk receptors of ecological risk assessment change constantly, the former from chemical pollutants caused by industrialization to global climate change, and components of ecosystems [17,18], and the latter from human health to natural environment ecosystems returning to human society [20–22]. The risk sources and risk receptors are diversified and complicated, and the ultimate target of ecological risk assessment is human society [23]. In the context of rapid urbanization, there is an increasing recognition of the interactions of the interplay of factors that might affect the risk faced by ecosystems, such as urbanization, industrialization, global climate change, land use change, landscape change, etc. [24]. Such interactions have indicated that unilateral risk management is unlikely to be useful in the management of complex systems [25]. The intertwined ecological risk source and risk receptor make the ecological risk management not only be done by a department, but also the structure of the ecological risk assessment provides a common framework for each department. This allows multiple stakeholders, regulatory organizations,

and scientists to agree on inherent difficulties of managing complex systems, which also suggests that risk assessment could be an important tool for multi-scale environmental management and decision making.

There are two major ecological risk assessment methods. One method reflects a relationship between regional risk receptor and risk source accurately, expresses regional risk source clearly, and constructs a relevant model to analyze regional ecological risks [17–20]. For example, Solomon et al. [26] analyzed the threat of atrazine in surface waters in North America, found important pollution factors for surface water, and promoted the healthy restoration of regional surface water quality. Beliaeff et al. [27] used an integrated biomarker response (IBR) to measure the response of mussel or fish tissues to chemical contaminants. Their results were compared to polycyclic aromatic hydrocarbons (PAH) or polychlorobiphenyls (PCB) levels measured in mussel or fish tissues. And they found that the IBR, as an indicator of environmental stress, appears to be a useful tool for scientists and managers in assessing ecological risk. Schmolke et al. [28] used an ecological model to comprehensively evaluate various negative effects of pesticides on human health. Correct analysis procedures of the relationship between risk sources and risk receptors are a basis for scientific and rational evaluation of regional ecological risks.

The other method is of ecological risk evaluation based on a landscape pattern [28,29]. The change in an ecological risk pattern of river basin landscape caused by human disturbance activity can be expressed as the response of risk receptors to risk sources (landscapes to human utilization activities such as deforestation, wetland reclamation and urban expansion, etc.) [29,30]. The method mainly uses landscape patterns or land use changes as an induction factor [28–30]. In some regions, due to lack of monitoring data of ecological environment, the ecological risk evaluation based on landscape patterns often makes a quantitative analysis and evaluation of influences of land use structures and types on a regional ecosystem by using the easily available data of land use types [30]. A landscape ecological risk assessment comprehensively assesses various potential ecological impacts and their cumulative effects, which is a supplement and extension of general ecological risk assessment. It explains the differences of ecological characteristics and risks among different landscape and assessment units through spatial heterogeneity and time series analysis. Kapustka et al. [31] suggested the ecological risk assessment of the area from a landscape ecology perspective. Ayre et al. [32] developed procedures that incorporate landscape features into the environmental management process and present a conceptual foundation for incorporating landscape ecology into the risk assessment process. Krajewski et al. [33] based on the map data of 140 years (1863–2013), using a landscape change index to analyze driving forces of forest resources change and forest transformation in a landscape park, and provided a theoretical and practical guidance for the protection of regional forest landscape resources. This kind of research also showed that the landscape change index was used as a way of monitoring landscape changes and therefore could be used for landscape ecological risk assessment [34]. A landscape pattern index is a true reflection of the regional landscape structure, layout, and function [35–37]. Under external disturbance, a landscape ecosystem tends to be broken or the stability is weakened, which can truly reflect the change in the landscape pattern index. Therefore, analyzing the landscape change index and understanding the temporal and spatial variation characteristics within the landscape is conducive to regional ecological environmental protection and ecological risk weakness. Given the important indicative significance of the landscape index, and the ease of operation and scientific rationality of the landscape ecological risk assessment, the method of landscape ecological risk assessment was adopted in this study.

An ecological risk assessment research area selected should reflect the centralized distribution of current ecological risk crisis areas. At present, the area of ecological risk assessment is concentrated in urban areas, river basins, coastal zones, etc. [38–40]. For example, Urrutiaaoyes et al. [41] took the industrial city of Monterrey in Mexico as a research object and analyzed the impact of exhaust gas and dust from automobiles and industries on urban ecology and human health. The impact on human health is a main content of current urban ecological risk assessment, including urban environmental

pollution, urban land use expansion, etc. There are increasing studies on river basins, coastal wetlands, etc. due to the intensification of human activities [42–44]. Compared with urban areas where human activities are obvious and intense, there are relatively few studies on watersheds and coastal areas where ecological environment is relatively fragile, which is consistent with the development trend of macro-ecology in recent years. These studies emphasize the impact of human activities on human ecosystems, and focus on the ecological environment-socio-economic complex [45]. As an important part of the natural ecological environment, ecologically fragile areas such as river basins, coastal wetlands, and desert oases not only provide material resources for the production and life of human society, but also play an unlimited role in regulating the climate, water conservation, and reserve resource storage. However, under constraints of rapid industrialization and urbanization, ecologically fragile areas are facing direct or indirect threats. The stability of their internal landscapes is affected; the ecosystems tend to be tight; and ecological risks are intensified. In this context, it is more realistic and practical to protect the ecological risk research in ecologically fragile areas.

River basins, as a relatively independent physical geographical unit, have a unique historical background and development process, and their internal components have frequent material, energy, and information exchange [46]. Leuven et al. [47] believed that rivers are dynamic and complex in structure, and that river basins ecological risk assessment requires long-term remote sensing image data to obtain temporal and spatial changes of multiple landscape types within the ecosystem [47]. At the same time, a basin ecological risk assessment needs to understand how to balance the interference of human activities on the landscape of the basin ecosystem, and the tolerance and resilience of the ecosystem itself. Land use, as a manifestation of human action on the landscape, embodies the interference of human activities on the regional landscape. Therefore, understanding the temporal and spatial patterns of land use within the basin and its changing processes can help understand the changes in different landscape structures and functions. It is also important to guide the comprehensive management of river basin ecological environment and the formulation of sustainable development plans. In river basins with high intensity of human activities and frequent land use transfers, land use changes are manifest due to the impacts of human activities on ecosystem in a natural environment, which shows evident regional and accumulation characteristics. Such impacts can be acted on composition and structure of an ecosystem directly, such as landscape type and ecological functions [48,49]. Therefore, the ecological risk assessment of negative impacts of external factors on ecosystem is scientific and feasible to some extents in river basins.

The Yongjiang River Basin locates at southeastern coastal zones with high economic development in China. Currently, studies on the Yongjiang River Basin mainly focus on river pollution [50]. Land use in Yongjiang River Basin is transferred greatly with the rapid economic development, which further destroys the stability of landscape pattern in the basin. In addition, land use changes have caused certain negative impacts on biodiversity and ecosystem service functions in the region. Under the background of implementing the most strict ecological environmental protection system, understanding influences of human activities on land use types and evaluating negative effects of regional land use changes on ecosystem are more beneficial to governing and protection of ecological environment in the basin. Therefore, in this study, we proposed to study land use changes and ecological risk assessment in Yongjiang River Basin in Zhejiang Province, China. Specifically, the research objectives include (1) analyzing land utilized degree in the Yongjiang River Basin based on land use changes; (2) given adverse impacts of regional land use changes on ecological environment, constructing an ecological risk index to evaluate landscape ecological risk of land use changes; and (3) according to ecosystem risk assessment results, providing a theoretical and practical guidance for regional reasonable development, protection of land resources, and maintaining stability of the ecological environment [51].

## 2. Materials and Methods

### 2.1. Study Area

The Yongjiang River Basin locates in southeastern coastal regions of China, northeastern region of Zhejiang Province, and the Sanjiang Plain in the eastern Ningshao Plain (Figure 1). It is between 29°24′ N–30°49′ N and 120°49′ E–120°56′ E. It covers an area of about 3788 km² and it is 105 km long. The Yongjiang River is one of the most important rivers in Ningbo and it includes Fenghua River and Yuyao River at the Sanjiang River estuary in Ningbo District. The Yongjiang River Basin is in the subtropical monsoon climate zone, where it has high terrains in the southwestern area and low terrains in the northeastern area. Most administrative regions of the Yongjiang River Basin are in Jiangbei District, Jinzhou District, Haishu District, Zhenhai District, Fenghua District, and Yuyao District of Ningbo City, while a small administrative region is in Beilun District, Cixi City, and Ninghai County. The study area enjoys advantageous locational conditions and convenient traffic. The Yongjiang River Basin promotes the rapid social and economic development significantly in Ningbo City. However, the land use types and structures were changed greatly during the rapid industrialization and urbanization, which caused violent changes in landscape patterns in the basin and intensified ecological risks in the region.

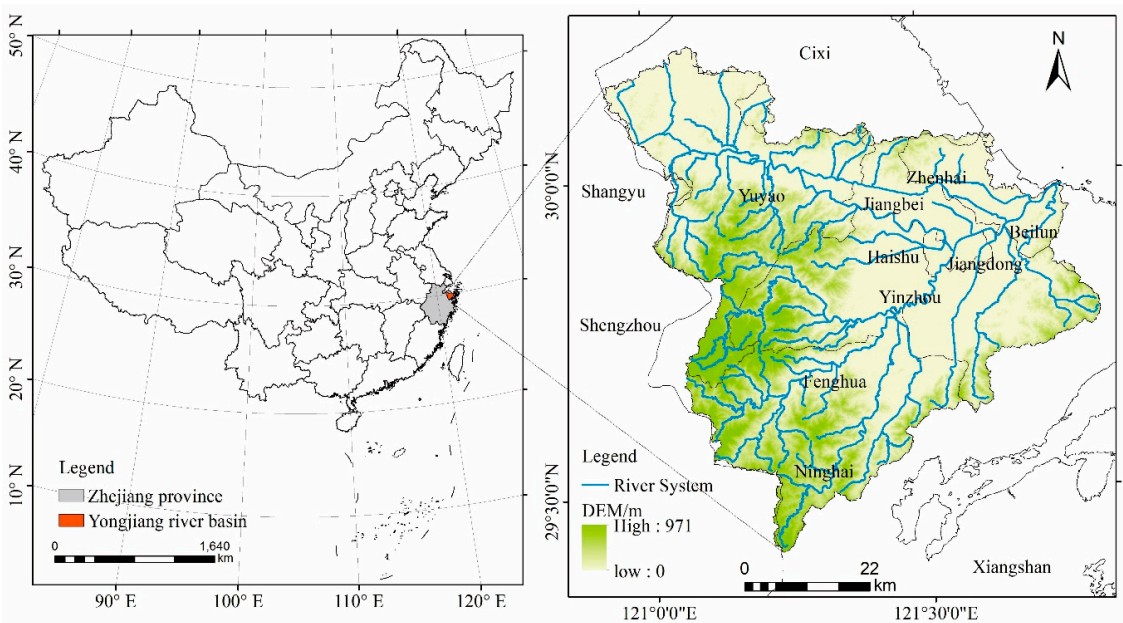

**Figure 1.** The geographical location of the study area.

### 2.2. Data Sets and Processing

All data used in this study were land use data from the Geographical Monitoring Cloud Platform (http://www.dsac.cn/) in a scale of 1:100,000 in Zhejiang Province in 1980, 1990, 1995, 2000, 2005, 2010, and 2015. All necessary research data could be gained from overlapping of administrative vector boundaries in the study area. According to the national land resource classification system, natural ecological background in the Yongjiang River Basin, and actual demands in this study, a total of 8 land use types were divided, including Cultivated Land, Forest Land, Grassland, Water Areas, Construction Land, Unused Land, Intertidal Zone, and Sea Areas covering the basin boundaries [52,53]. Intertidal Zone mainly refers to the tide invasion zone between the high tide level and low tide level in coastal regions. Definitions of other land use types are introduced in reference [52].

### 2.3. Landscape Change Index

The land use change characteristics in the study area were analyzed by calculating the landscape change index (LCI) of each period. In order to calculate the landscape change index, it was necessary to get the change in the percentage share of the area covered by each land cover type [33]. It was calculated as follows:

$$CA_i = 100 \times (A_{t+1} - A_t)/TA \tag{1}$$

where $CA_i$ represents the percentage share of a land cover type to all land cover types in the study area (%); $A_{t+1}$ represents the area covered with each type of land cover during the time interval t + 1 (km$^2$); $A_t$ represents the area covered with each type of land cover during the time interval t (km$^2$); and $TA$ represents the total research area (km$^2$).

The landscape change index (*LCI*) was used to reflect the overall change in land cover in the study area. The LCI is defined as the absolute values of change in land cover types that have the greatest impact on the shape of the landscape, assuming that both increases and decreases in these values cause corresponding changes in the landscape [54]. Summing the absolute values of change in each land cover type essentially doubles the index, so the LCI includes a factor of one-half to reflect the actual level of change. LCI was calculated with

$$LCI_t = \frac{1}{2} \times \sum_{i=1}^{n} |CA_i| \tag{2}$$

where $LCI_t$ represents the landscape change index in each time interval; and $|CA_i|$ represents the absolute value of change in percentage share of the areas covered by each land cover type in relation to the total research area [33].

### 2.4. Land Use Transfer Analysis

The land use transfer matrix in different periods was calculated by the Markov model, through which the land use transfer direction and areas in the study area in 35 years were disclosed. The calculation formula is shown in reference [55]. According to the calculated land use transfer matrix, proportions of net losses and net gains of land use types at different stages were calculated [55]:

$$P_{loss(i)j} = \frac{(P_{j,i} - P_{i,j})}{(P_{i.} - P_{.i})} \times 100\%, \ i \neq j$$
$$P_{gain(i)j} = \frac{(P_{j,i} - P_{i,j})}{(P_{i.} - P_{.i})} \times 100\%, \ i \neq j \tag{3}$$

where $P_{loss(i)j}$ and $P_{gain(i)j}$ are the proportions of net loss and net gains of land use type *i* during transfer into land use type *j* in the matrix row. They are contribution rates of variations. $P_{i,j}$ and $P_{j,i}$ are the specific numerical values in the matrix.

### 2.5. Land Utilized Degree

With references to previous studies, e.g., [56], the land utilized intensity was quantized. The regional land utilized intensity was expressed by the index of land utilized degree:

$$L = 100 \times \sum_{i=1}^{n} A_i \times C_i \tag{4}$$

where *L* is the comprehensive index of regional land utilized degree; $A_i$ is the land use index of grade *i*; $C_i$ is the proportion of land utilized degree *i*; and *n* is the number of grades of land utilized degree. Based on previous experiences, e.g., [56], and research demands, grades of land utilized degrees have to be divided (Table 1).

| Type | Unused Land Grade | Forest, Grass and Water Land Grade | Agricultural Land Grade | Urban Settlement Land Grade |
|---|---|---|---|---|
| Land use type | Unused land, Beach | Woodland, waters | Cultivated land | Construction land |
| Graded index | 1 | 2 | 3 | 4 |

*2.6. Ecological Risk Assessment*

Changes of landscape structures and functions in the Yongjiang River Basin caused by land use transfer were used to reflect responses of risk receptors to human development activities [57,58]. Dynamic changes in a landscape pattern in the study area were reflected clearly through quantification of landscape indexes corresponding to the landscape index method. The landscape interference degree ($E_i$) and vulnerability degree ($F_i$) were selected to create the landscape loss index ($R_i$) to represent landscape loss upon impacts of the external environment. The formula of calculating $R_i$ is

$$R_i = E_i \times F_i \tag{5}$$

where $E_i$ is the impact of external environment on different landscapes. It can be expressed as

$$E_i = aC_i + bN_i + cD_i \tag{6}$$

where $C_i$ is the landscape fragmentation that reflects the degree of fragmentation of landscape in the basin and stability in the landscape. $N_i$ is the landscape isolation that shows isolation of landscape plaques and reflects discretion in the landscape. $D_i$ is the landscape advantage which shows advantages of a plaque in one landscape type and proves its importance in the landscape. $a$, $b$, $c$ are the weights of $C_i$, $N_i$, and $D_i$, which value 0.5, 0.3, and 0.2 with references to existing research results [57,58]. $F_i$ denotes the structural vulnerability of the landscape ecosystem. High value of $F_i$ implies the tendency of ecosystem toward poor safety. Based on existing results and expert consultation [58], all 8 land use types were graded according to resistance to external interferences: Unused Land = 8, Intertidal Zone = 7, Sea Areas = 6, Water Areas = 5, Cultivated Land = 4, Grassland = 3, Forest Land = 2, and Construction Land = 1. The $F_i$ was calculated through normalization.

According to the study area and previous studies [57–59], the fish net should be 2–5 times the average plaque area in the study area [59]. A 2 km × 2 km grid was created by using ARCGIS10.3 fish net tool, and the study area was divided into 1058 risk regions. Combining $R_i$ and areas of different sampling plots, the ecological risk index (ERI) was expressed as

$$ERI_i = \sum_{i=1}^{N} \frac{A_{ki}}{A_k} R_i \tag{7}$$

where $ERI_i$ is the ecological risk value of landscape type $i$; $A_{ki}$ is the area of type $i$ in the sampling plot $k$, and $A_k$ is the area of the sampling plot $k$ [59].

*2.7. Spatial Analysis Method of Ecological Risk*

The ecological risk value is given to the central point of risk regions in ARCGIS10.3. The ecological risk map of the study area at different stages was produced by Kriging interpolation under the spatial analysis module [58,59]. For better recognition of ecological risk changes of landscape and uniform evaluation in different periods, the ecological risk index value of risk regions was divided at an equal interval by using the relative index method [17]. The ecological risk value was divided into five grades at an interval of 0.03: low ecological risk zone ($ERI < 0.12$), relatively low ecological risk zone ($0.12 < ERI \leq 0.15$), middle ecological risk zone ($0.15 < ERI \leq 0.18$), relatively high ecological risk zone ($0.18 < ERI \leq 0.21$), and high ecological risk zone ($ERI > 0.21$).

## 3. Results and Analyses

In this section, a concise and precise description of the experimental results was provided. Their interpretation and brief analyses were conducted.

### 3.1. Characteristics of Land Use Changes

The land use types in the Yongjiang River Basin from 1980 to 2015 were sorted out (Figure 2): the land use types in the Yongjiang River Basin have undergone major changes in quantity and structure. The grassland decreased greatly in the first 10 years, and the change area was smaller in the last 25 years. The cultivated land visually slightly decreased before 1995, and the overall trend continued to decrease, decreasing by 553.51 km$^2$. The sea area remained basically unchanged in 2010 and before, and rapidly decreased to disappear in the last five years. Construction land continued to increase rapidly and the positive growth rate was obvious, from 3.91% in 1980 to 18.60% in 2015, a total increase of 556.63 km$^2$, with a growth rate of 376%. The area of forestland increased first and then decreased in 1995, and the overall change was small. The water area continued to increase, but the growth rate was uneven. The intertidal zone and unused land area changed little, and intertidal zone finally disappeared in years 2010–2015.

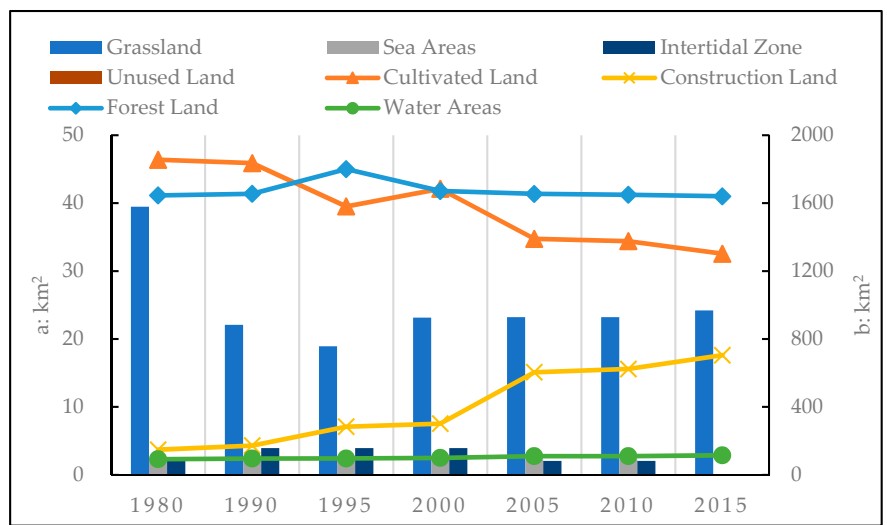

**Figure 2.** Land use change in the study area (Note: a: Grassland, Unused Land, Intertidal Zone and Sea Areas; b: Cultivated Land, Construction Land, Forest Land, Water Areas).

The CA value and the landscape change index level (LCI) in the seven time periods of the study area from 1980 to 2015 were calculated (Table 2). From the perspective of each land use type, the land use types changed little and the CA values were lower from 1980 to 1990. From 1990 to 1995, the change in cultivated land was prominent, and its area decreased greatly. The area of construction land and forestland increased. From 1995 to 2000, the area of forestland decreased greatly. From 2000 to 2005, the construction land increased sharply, with the highest CA value of 8.00%, followed by cultivated land, in which the CA value was −7.77%, indicating that the growth of construction land mainly comes from the occupation of cultivated land. The land use change was not obvious from 2005 to 2010. From 2010 to 2015, the area change of cultivated land and construction land was more prominent. Overall, from 1980 to 2015, in the change in land use type in the study area, the changes in cultivated land and construction land showed remarkably, in which the cultivated land area decreased sharply, and the construction land area increased rapidly. The CA values of the two types were similar, which were −14.60% and 14.70%, respectively. Per the change in LCI level, it mainly shows the fluctuation state, and tends to increase in general. The LCI level is the largest in years 2000–2005, while it is the smallest in 2005–2010, and the maximum index is 16.54 times the minimum index. Among different time periods, LCI was

relatively large in years 1995–2000 and 2000–2005. During the same time periods, land use changes were prominent, and various land use conversions were complicated. But in the other time periods, the land use changes were small. In summary, from 1980 to 2015, the land use in the Yongjiang River Basin changed significantly, and land development and utilization were active.

**Table 2.** Dynamics of changes in land covers in the study area between 1980 and 2015.

| Land Use Types | Indicator | 1980–1990 | 1990–1995 | 1995–2000 | 2000–2005 | 2005–2010 | 2010–2015 | 1980–2015 |
|---|---|---|---|---|---|---|---|---|
| Grassland | | −0.46 | −0.08 | 0.11 | 0.00 | 0.00 | 0.03 | −0.40 |
| Cultivated land | | −0.53 | −6.73 | 2.73 | −7.77 | −0.36 | −1.96 | −14.61 |
| Sea areas | | 0.00 | 0 | 0 | 0.00 | 0 | −0.06 | −0.06 |
| Construction land | CA | 0.64 | 2.94 | 0.47 | 8.00 | 0.50 | 2.15 | 14.70 |
| Forest land | (%) | 0.25 | 3.84 | −3.41 | −0.45 | −0.13 | −0.23 | −0.13 |
| Water areas | | 0.09 | 0.02 | 0.09 | 0.27 | −0.01 | 0.13 | 0.60 |
| Intertidal zone | | 0.05 | 0.00 | 0.00 | −0.05 | 0 | −0.05 | −0.06 |
| Unused land | | −0.04 | 0.01 | 0.00 | 0.00 | 0 | 0.00 | −0.03 |
| | LCI | 1.02 | 6.81 | 3.41 | 8.27 | 0.50 | 2.31 | 15.30 |

0.00 is the result of rounding.

The main characteristics of land use transfer in the study area were introduced (Table 3; Table 4). In general, Cultivated Land was mainly transferred to Construction Land, Forest Land, and Water Areas. Sea Areas, Forest Land, and Unused Land were mainly transferred to Construction Land. Grassland was mainly transferred to Forest Land. Land use type transfer in the study area during the study period was analyzed as follows. The most land use transfer was from Cultivated Land to Construction Land. Specifically, 89.36% of Cultivated Land was transferred to Construction Land and the transfer area was 521.90 $km^2$. The 93.03% of Grassland was transferred to Forest Land and Sea Areas was mainly transferred to Construction Land and Grassland. During the urbanization and industrialization, Construction Land had greater transfer-in sources than those of transfer-out areas. Specifically, Cultivated Land and Forest Land were two major types of transfer-in sources of Construction Land, accounting for 92.61% and 6.29%, respectively. Due to policies of returning the grain plots to forestry, Forest Land was mainly transferred to Construction Land and Cultivated Land. Water Areas were mainly transferred to Cultivated Land and Construction Land, while Intertidal Zone was mainly transferred to Water Areas, and Unused Land was mainly transferred to Construction Land. Sea Areas and Intertidal Zone had a small area in early period, which, however, were all transferred into other land use types in the late study period. During the 35 years, the land use transfer among different types became increasingly complicated with the increase of economic activity intensity in the Yongjiang River Basin.

**Table 3.** Transition matrix of each land use type in Yongjiang River Basin during 1980–2015 ($km^2$).

| Land Use Type | Grassland | Cultivated Land | Construction Land | Forest Land | Water Areas | Unused Land | Total |
|---|---|---|---|---|---|---|---|
| Grassland | 9.90 | 0.71 | 1.18 | 27.50 | 0.05 | 0.12 | 39.46 |
| Cultivated land | 0.93 | 1271.57 | 521.90 | 37.47 | 23.73 | 0.00[1] | 1855.60 |
| Sea areas | 0.81 | 0.02 | 1.50 | 0.01 | 0.00 | 0 | 2.34 |
| Construction land | 0.01 | 5.44 | 141.11 | 0.85 | 0.63 | 0 | 148.04 |
| Forest land | 12.53 | 22.12 | 35.46 | 1573.67 | 1.17 | 0.84 | 1645.79 |
| Water areas | 0.01 | 2.15 | 1.60 | 1.07 | 87.46 | 0 | 92.28 |
| Intertidal zone | 0 | 0.02 | 0.70 | 0 | 1.47 | 0 | 2.19 |
| Unused land | 0.02 | 0.06 | 1.21 | 0.16 | 0.57 | 0.16 | 2.19 |
| Total | 24.20 | 1302.09 | 704.67 | 1640.73 | 115.08 | 1.13 | 3787.89 |

0.00 is the result of rounding.

**Table 4.** Internal conversions between land use types in Yongjiang River Basin during 1980–2015 (km$^2$).

| Land Use Type | Interm Increase or Decrease Rate % | Conversion Type | Contribution Rate % | Conversion Type | Contribution Rate % | Conversion Type | Contribution Rate % | Conversion Type | Contribution Rate % | Conversion Type | Contribution Rate % |
|---|---|---|---|---|---|---|---|---|---|---|---|
| Grassland | −38.68 | Cultivated land | 2.39 | Construction land | 4.01 | Forest land | 93.03 | Water areas | 0.17 | Unused land | 0.41 |
| Cultivated land | −29.83 | Grassland | 0.16 | Construction land | 89.36 | Forest land | 6.42 | Water areas | 4.06 | Unused land | 0.00[1] |
| Sea areas | −100.00 | Grassland | 34.72 | Cultivated land | 0.74 | Construction land | 64.27 | Forest land | 0.24 | Water areas | 0.03 |
| Construction land | 376.00 | Grassland | 0.12 | Cultivated land | 78.53 | Forest land | 12.28 | Water areas | 9.08 | Unused land | 0.00[1] |
| Forest land | −0.31 | Grassland | 17.37 | Cultivated land | 30.67 | Construction land | 49.16 | Water areas | 1.63 | Unused land | 1.16 |
| Water areas | 24.70 | Grassland | 0.12 | Cultivated land | 44.55 | Construction land | 33.21 | Forest land | 22.12 | Unused land | 0.00[1] |
| Intertidal zone | −100.00 | Grassland | 0.00[1] | Cultivated land | 0.80 | Construction land | 32.06 | Forest land | 0.00[1] | Water areas | 67.14 |
| Unused land | −48.56 | Grassland | 1.06 | Cultivated land | 3.12 | Construction land | 59.66 | Forest land | 7.90 | Water areas | 28.27 |

0.00 is the result of rounding.

Spatially (Figure 3), Cultivated Land was extensively transferred into Construction Land in the study area, which was mainly in flat regions with good geographical positions. There is a high land utilized degree and big land use demands close to urban areas with strong human activities. With the policies of returning Cultivated Land into Forest Land, Cultivated Land was transferred into Forest Land in some regions, mainly in regions with large topographic relief that were unsuitable for cultivation like mountainous and hilly regions. Cultivated Land transferring into Water Areas was mainly in East China. Land use type changed as a response to the emerging of coastal aquaculture. Faced with increasing population and increasing pressure of food security, Forest Land was transferred into Cultivated Land mainly in relatively flat regions that are suitable for cultivation. Forest Land was transferred to Construction Land mainly in suburbs of cities that were mainly in the range of city expansion and influenced by economic development level and speed significantly. Grassland was transferred into Forest Land mainly in gentle mountainous and hilly regions which were in Midwest regions of the study area.

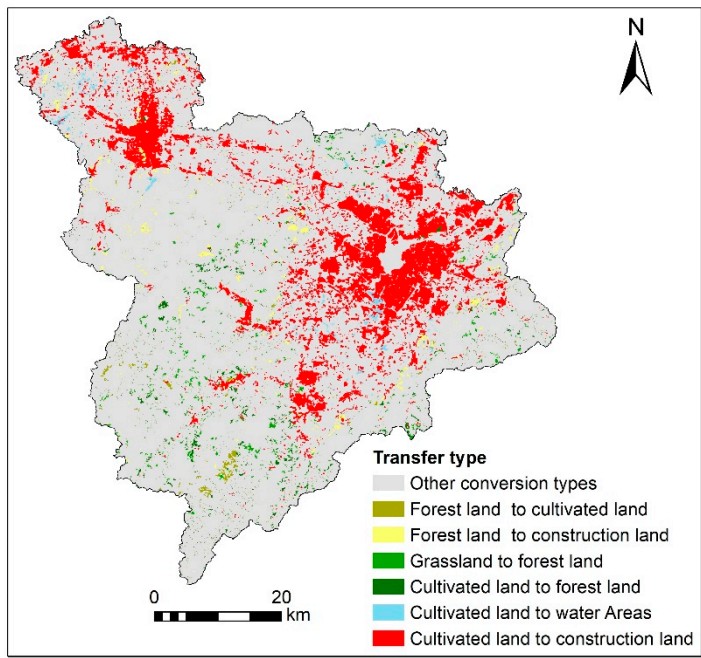

**Figure 3.** Change patterns of land use in Yongjiang River Basin during 1980–2015.

### 3.2. Land Utilized Degree

Land utilized degree in the Yongjiang River Basin was calculated (Figure 4) and it was divided in an equal interval of 60 as the natural breakpoint. The evaluation results were divided into five grades: weak ($100 < L \leq 160$), relatively weak ($160 < L \leq 220$), middle ($220 < L \leq 280$), relatively strong ($280 < L \leq 340$), and strong ($340 < L \leq 400$). Spatial variation of land utilized degree in the study area was reflected by spatial vectorization. Temporally, the land utilized degree in the study area generally increased from 1980–2015 due to the rapid economic development, manifested by increased forms and intensity of human activities. During 1990–1995, the land utilized degree declined slightly, but it increased quickly by 8.27 during 2000–2005.

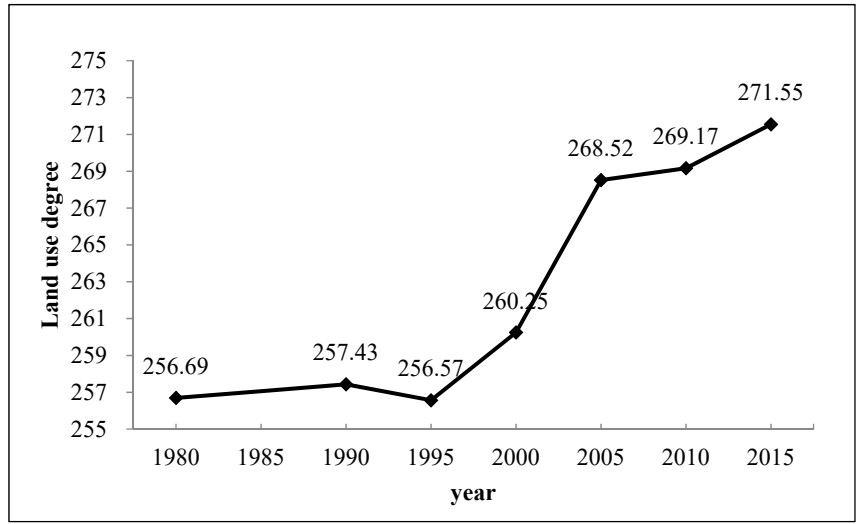

**Figure 4.** Land utilized degree in Yongjiang River Basin during 1980–2015.

Spatially (Figure 5), there is an evident regional difference in land utilized degree in the past 35 years. The land utilized degree in the northern study area was higher compared with that in southern regions, and higher in eastern regions than in western regions. In the study area, the land utilized degree was generally high. Areas with middle or higher land utilized degree increased, while the proportion of areas with strong land utilized degree was increased quickly at a stable rate. In 2005, the area of high land use intensity increased mostly, which was also related to the substantial increase in construction land area at this time. The area with relatively strong land utilized degree was decreased continuously in general, while the area with middle land utilized degree changed slightly. The areas with relatively weak and weak land utilized degree continued to decrease and reached the minimum in 2015. Due to limitations on land use in mountainous and hilly regions in southwestern areas, the land utilized degree in eastern and northern regions of the study area increased gradually. The eastern and northern regions in the Yongjiang River Basin were influenced significantly by human activities due to the application of chemical fertilizers in Cultivated Land and pollutions by household wastes and solid wastes at settlements, factories, and enterprises. There regions suffered increasing pressures from production and daily life.

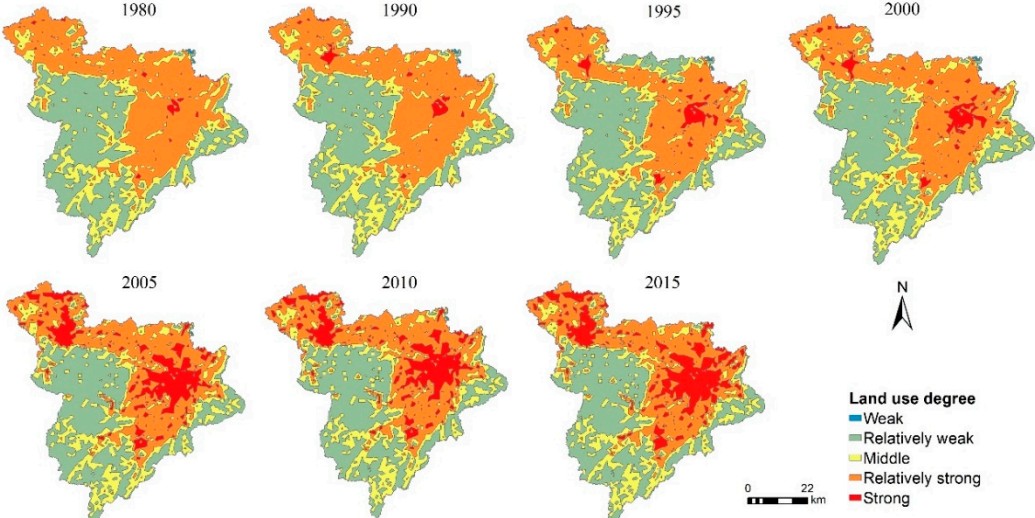

**Figure 5.** Spatial distribution of land utilized degree in Yongjiang River Basin during 1980–2015.

### 3.3. Ecological Risk Assessment

3.3.1. Spatiotemporal Variation Characteristics of Ecological Risks

The Yongjiang River Basin is in southeastern coastal regions in China with a rapid economic development. Land use types are changing greatly and the land utilized degree increases gradually. Variations of landscape patterns caused by land use changes are affecting the ecosystem in the basin. The continuous accumulation of these variations thereby threats stability and serviceability of the ecosystem. Based on landscape pattern changes caused by land use changes, an ecological risk assessment model was introduced to reflect spatiotemporal variation characteristics of land use ecological risks (Figure 6; Figure 7). It was found that the ecological risk in the Yongjiang River Basin during 1990–2015 fluctuated violently in time and space.

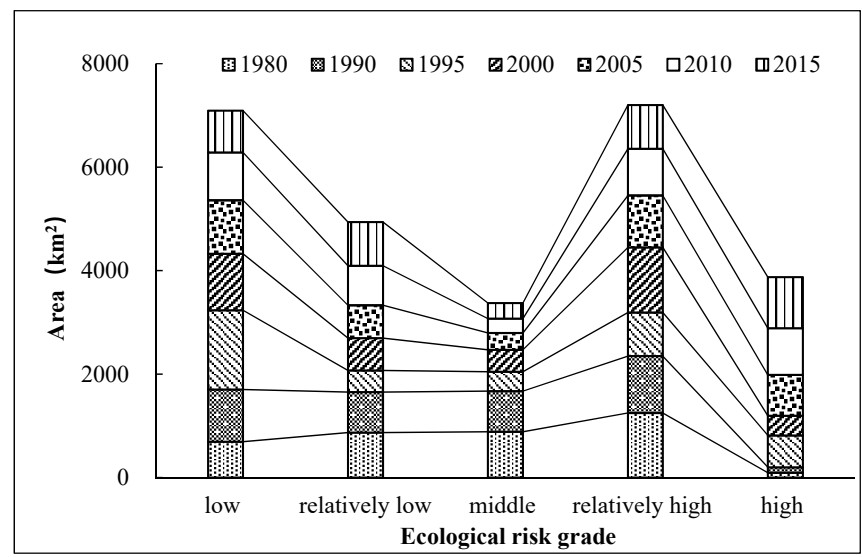

**Figure 6.** Area changes of ecological risk area in Yongjiang River Basin during 1980–2015.

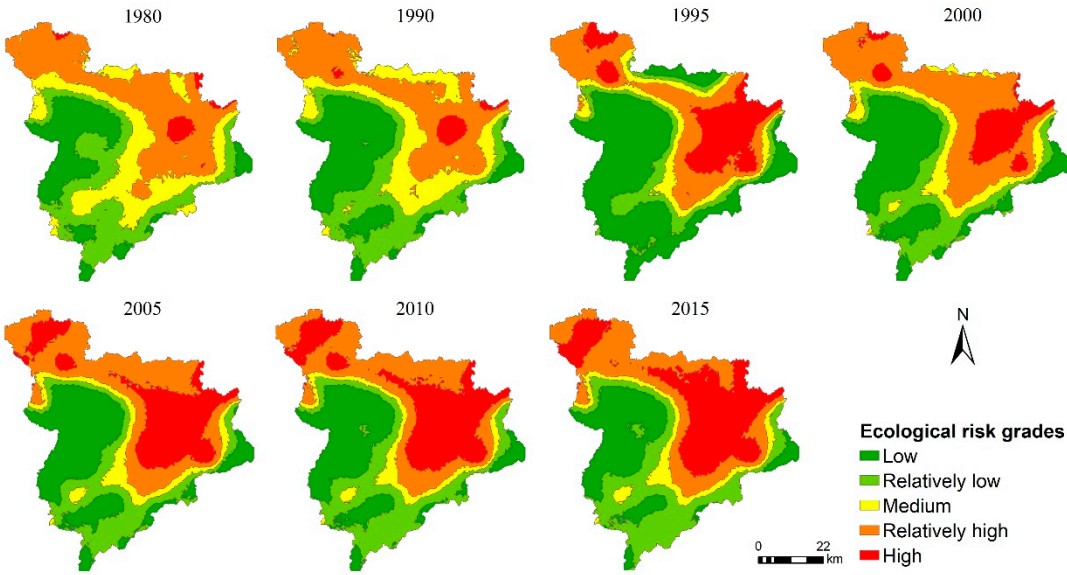

**Figure 7.** Spatial distribution of ecological risk in Yongjiang River Basin during 1980–2015.

Temporally, the ecological risk in the Yongjiang River Basin generally increased in a fluctuating manner during 1990–2015. The area of the low ecological risk region increased firstly and then decreased and it was increased by 113.49 km² in the late study period. The area of the relatively low

ecological risk region decreased by 869.24 km$^2$ in 1980 to 417.06 km$^2$ in 1995, and then increased to 841.99 km$^2$ in 2015. The area with the middle ecological risk region decreased by 66.06%. The area of the relatively high ecological risk region declined slightly by 402.90 km$^2$ in a fluctuating manner. The area of the high ecological risk region increased in a fluctuating manner. The proportion of area of the high ecological risk region to the total area of the basin increased by 893.77 km$^2$ from 2.46% in the early period to 26.10% in the late study period. During 1980–1990, the high ecological risk region took the dominant role. The low ecological risk region occupied the largest area in 1995 and during 2005–2010, but the growth rate of the high ecological risk region occupied the dominant role in 2015. The rapid growth of area of the high ecological risk region implied the profound impacts of external interference factors on the natural environmental ecosystem during land use changes in the basin. Moreover, the sustainability and stability of ecosystem in the region were destroyed in this period. More attentions shall be paid to protection of ecological environment, reasonable exploitation and allocation of land resources.

Spatially, the ecological risk in the study area fluctuated greatly. Low and relatively low ecological risk regions concentrate in western and southern regions in the Yongjiang River Basin, while high and relatively high ecological risk regions concentrate in northern and eastern regions of the basin, especially in northeastern regions. The Yongjiang River Basin converges to the East China Sea at Zhenhai Estuary in Ningbo. The eastern region is in the downstream of the Yongjiang River Basin. The northern and eastern regions are in downtown, Yuyao and other urban regions in Ningbo, which have high urbanization level, frequent human activities, and strong land utilization, thus increasing ecological risk in the region. Driven by a rapid economic development, Cultivated Land and Water Areas with low ecological risk were transferred to Construction Land with high ecological risk. Western and southern regions of the basin are in upstream and they are ecological conservation areas that are mainly in mountainous and hilly regions with a large terrain relief. The area of Forest Land was large and stable, which was difficult to be transferred into land use type with high ecological risk at a large scale. The economic development in western and southern regions relatively lagged behind compared with the eastern coastal regions, resulting in the low land utilized degree in these regions. Moreover, the grade of ecological risk in western and southern regions was lower than that in eastern and northern regions. With respect to spatial distribution, the ecological risk in the Yongjiang River Basin was high in northeastern regions and low in southwestern regions. High ecological risk regions expanded gradually from the areas with a lower grade. The middle ecological risk region was a stripped region that run from the east to the west. It was mainly distributed along Water Areas.

3.3.2. Ecological Risk Transfer

Transfer of ecological risk regions in different periods was reflected by constructing a transfer matrix (Table 5). It was found that the area of ecological risk grade transferred from low to high was 4.19 times the area of ecological risk grade transferred from high to low, whereas the high-grade ecological risk region was expanded. Low, relatively low, middle, and relative high ecological risk regions were mainly transferred to the high grade of ecological risk regions. The transfer area from relatively high ecological risk region to high ecological risk region was the highest, reaching 813.07 km$^2$. The transfer probability of middle ecological risk region was the highest (71.51%) and the transfer area was 631.72 km$^2$. High ecological risk region had the second highest transfer area and probability, which were 813.10 km$^2$ and 65.17%, respectively. The transfer area and probability of low ecological risk region were 258.26 km$^2$ and 29.71%, respectively. The hotspot transfer regions were mainly in northern and eastern regions. Regions close to the center of Yuyao City, downtown regions of Ningbo City and center of Fenghua City witnessed great impacts of human activities and had high land utilized degree, thus making the ecological risk transfer complicated. Ecological risk was transferred from the low grade to the high grade gradually.

Mutual transfer and changes of six ecological risk regions were analyzed (Table 6). A total of 15 transfer directions among six ecological risk regions were recognized and the transfer was

complicated [30]. In the five time periods, ecological risk transfer from high grade to low grade mainly includes seven directions: relatively low–low, middle–low, middle–relatively low, relatively high–low, relatively high–relatively low, relatively high–middle, and high–relatively high. Ecological risk transfer from low grade to high grade mainly covers eight directions: low–relatively low, low–middle, low–relatively high, relatively low–middle, relatively low–relatively high, middle–relatively high, middle–high and relatively high–high. The area of ecological risk transferring from middle grade to relatively high grade and from relatively high grade to high grade was large, reaching 698.76 km$^2$ and 1209.62 km$^2$, respectively. In the five study periods, areas of ecological risk transferring from low grade to high grade were 98.51 km$^2$, 897.60 km$^2$, 640.36 km$^2$, 685.57 km$^2$, 247.76 km$^2$, and 324.66 km$^2$, showing an increasing trade in the study period. During 1990–2000, the area of ecological risk transferring from low grade to high grade was large, while the high ecological risk region diverged. Areas of ecological risk transferring from the high grade to the low grade were 762.12 km$^2$, 693.31 km$^2$, 268.77 km$^2$, 17.96 km$^2$, 26.95 km$^2$, and 59.56 km$^2$, showing a decreasing trend. This reflected that the area of low ecological risk region shrank significantly, but the area of high ecological risk region was expanded, which intensified the ecological risk in the basin. The area of high ecological risk region increased quickly, which disclosed the urgency to relieve a contradiction between economic development and ecosystem.

**Table 5.** Transition matrix of land use ecological risk area during 1980–2015 (km$^2$).

| 2015 <br> 1980 | Low | Relatively Low | Middle | Relatively High | High | Total |
|---|---|---|---|---|---|---|
| low | 600.09 | 94.22 | 0 | 0 | 0 | 694.31 |
| relatively low | 209.45 | 610.98 | 48.81 | 0.01 | 0 | 869.24 |
| middle | 0.05 | 135.88 | 251.63 | 411.04 | 84.75 | 883.35 |
| relatively high | 0 | 0.03 | 0 | 434.54 | 813.07 | 1247.63 |
| high | 0 | 0 | 0 | 1.42 | 91.93 | 93.35 |
| Total | 809.58 | 841.11 | 300.43 | 847.00 | 989.76 | 3787.89 |

**Table 6.** Analysis of the direction of ecological risk transfer during 1980–2015 (km$^2$).

| Direction of Ecological Risk Grade Transfer | 1980–1990 | 1990–1995 | 1995–2000 | 2000–2005 | 2005–2010 | 2010–2015 |
|---|---|---|---|---|---|---|
| From low to relatively low | 32.74 | 0.00 | 351.32 | 61.30 | 108.28 | 118.47 |
| From low to middle | 0.00 | 0.00 | 29.89 | 0.00 | 0.00 | 0.00 |
| From low to relatively high | 0.00 | 0.00 | 58.80 | 0.00 | 0.00 | 0.00 |
| From relatively low to low | 343.89 | 432.87 | 0.01 | 2.62 | 0.21 | 0.18 |
| From relatively low to middle | 2.36 | 23.45 | 92.32 | 59.93 | 0.74 | 42.23 |
| From relatively low to relatively high | 0.00 | 0.00 | 49.71 | 0.00 | 0.00 | 0.00 |
| From middle to low | 3.60 | 86.04 | 0.00 | 0.00 | 0.00 | 0.00 |
| From middle to relatively low | 230.34 | 61.14 | 0.15 | 7.68 | 26.43 | 0.17 |
| From middle to relatively high | 31.89 | 362.76 | 107.30 | 152.02 | 20.65 | 24.14 |
| From middle to high | 0.00 | 4.26 | 0.00 | 0.00 | 0.00 | 0.00 |
| From relatively high to low | 0.00 | 6.83 | 0.00 | 0.00 | 0.00 | 0.00 |
| From relatively high to relatively low | 0.03 | 27.25 | 0.00 | 0.00 | 0.00 | 0.00 |
| From relatively high to middle | 164.95 | 79.19 | 32.64 | 3.00 | 0.14 | 1.80 |
| From relatively high to high | 31.52 | 507.12 | 0.74 | 412.32 | 118.09 | 139.83 |
| From high to relatively high | 19.31 | 0.00 | 235.96 | 4.66 | 0.17 | 57.40 |

## 4. Discussion

### 4.1. Effects of Land Use Changes on Ecological Risk in the Study Area

Land is the macroscopic representation of surface landscape. Land use is one of the most intuitive expression forms of humans developing and using the physic-geographical environment [49,50]. Structural or pattern changes of land are highly correlated with spatial and temporal distribution

and dynamics of landscape ecological risk [60]. Land use changes not only influence single ecological elements like soil environment, atmospheric environment, and water environment, but also affect maintaining the comprehensive ecological process and environmental evolution in a basin, thus threatening ecological health and ecological security in the study area. Influences of number of landscape elements, functions and combination modes on ecological risks were analyzed by landscape ecological risk evaluation based on spatial pattern. Land use changes influence the normal development and operation of landscape structures and functions to different extents through landscape patterns and landscape elements of the regional ecosystem. Continuous accumulation of these influences can affect stability and serviceability of the ecosystem in the study area [61].

Ecological risk of land use in the Yongjiang River Basin was increasing gradually. On the one hand, the land use structure in the study area changed greatly from 1980 to 2015. Given the background of rapid urbanization, the urban area of Ningbo and Yuyao were expanded quickly in the basin. Cultivated Land and Forest Land were transferred to Construction Land at a large scale. In the same way, other land use types were transferred to Construction Land. Due to the dramatic increase of Construction Land, the plaque area of a landscape in the basin tended to fragmented. In particular, embezzlement of Cultivated Land for urbanization as well as construction of infrastructure (e.g., transportation and network circuit) destroyed stability in landscape in Cultivated Land, Forest Land, and Grassland, which intensified the regional ecological risk. On the other hand, the land use intensity in the basin increased quickly in the 35 years. Land use intensity reflected the degree of land utilized by human activities, and land use intensity in the basin was increased significantly, indicating the strong interference of human activities to land use intensity. The Yongjiang River Basin is in the developed southeastern coastal regions in China, with a long history of land use development. The basin serves for multiple functions, such as watercourse transportation, agricultural irrigation, water conservation, water consumption for urban production and life, tourism, etc. Terrain in the economically developed Ningbo City at downstream of the Yongjiang River Basin and Yuyao City that the Fenghua River passes through was flat. Human activities can disturb land use intensity greatly. Construction Land was dense and tended to be expanded, while the area of Cultivated Land decreased. Water Areas and Intertidal Zone in coastal regions were reclaimed for cultivation, showing complicated land use types and changes as well as intensifying ecological risks in the region. The western and southern regions were used as the upstream of Yongjiang River Basin, which were mainly occupied by mountainous regions. There was an extensive distribution of Forest Land. Moreover, the upstream was used as the waterhead conservation area and human activities disturbed the land utilized degree slightly. The land utilized degree and ecological risk in the upstream were lower than those in eastern regions.

## 4.2. Importance of Ecological Risk Assessment in the Basin

The administrative region is used as the evaluation unit, which is beneficial for decision-makers to formulate different policies for ecological risk control in different administrative regions. It is an ideal unit for risk assessment in regions with strong human interferences and high urgency of ecological recovery. Relatively speaking, a surface natural boundary, such as watershed, is used as the boundary of the evaluation unit, representing natural landscape distribution under natural conditions. In an administrative region, calculating ecological risk can assure integrity of natural element structure and process in one unit. This demonstrates that there is a similar risk source and natural environment in a small river basin. There is a great difference among different small basins in term of risk sources and natural environment. In other words, intra-region homogeneity and inter-region homogeneity of ecological risks are mostly significant in representation of the basin.

Currently, there are many studies on ecological risk assessment of land use in regions with strong human activities. However, there are few studies on ecological risk assessment in natural land areas, such as river basin, forest, and grassland. Meanwhile, further deep studies on ecological risk assessment in natural terrains (e.g., river basin, coastal zone, wetland, ecotone of agriculture–animal husbandry, and desert oasis) with vulnerable ecology and strong responses to global changes are

still needed. The overall landscape pattern of these regions is relatively fragmented and has poor stability and recovery ability. The overall landscape patterns of these regions are changed quickly and obviously under the disturbances of human activities and natural factors. With considerations to landscape patterns and dynamic measurement of ecosystem health conditions, determining species habitat and stresses on environmental factors has important ecological significance. These can guide practices for regional repair of ecosystem and coordinated development between social economy and natural environment.

## 5. Conclusions

The ecological risk assessment method was conducted based on the information of landscape patterns and land use type changes to characterize the response of regional ecosystems to human activities (urban expansion, vegetation destruction, water pollution, cultivated land encroachment) and global climate change. This method has a better environmental indicative significance, especially in river basins where the ecological environment is relatively fragile. Land use change affects regional ecosystems through different changes and their mutual influences, and accumulates to threaten the stability and sustainability of ecosystems. Therefore, based on the changes in area, structure, and function of land use types in Yongjiang River Basin during 1980–2015, spatiotemporal variation characteristics of ecological risk in the study area were evaluated. Some major conclusions could be drawn from the experimental results:

(1) Grassland, Forest Land, Cultivated Land, Sea Areas, Intertidal Zone, and Unused Land were decreased in the Yongjiang River Basin, while Construction Land and Water Areas were increased continuously. Cultivated Land was mainly transferred into Construction Land, Forest Land, and Water Areas, while Sea Areas, Forest Land, and Unused Land were mainly transferred to Construction Land. Grassland was mainly transferred to Forest Land. Land use transfer became increasingly complicated.

(2) Land utilized degree in Yongjiang River Basin was increased by 57.89% in the late study period. There is a big regional differentiation. Influenced by terrain conditions, geographical location, and economic development, the land utilized degree in northern regions was higher than that in southern regions and the land utilized degree in eastern regions was higher than that in western regions.

(3) Ecological risk in the Yongjiang River Basin generally increased in a fluctuating manner. The area of low ecological risk region increased firstly and then decreased, while the area of relatively low ecological risk region showed an opposite variation law. The area of middle ecological risk region decreased continuously, while the area of relatively high ecological risk region decreased slightly in the fluctuating manner. The area of high ecological risk region was increased by 893.96 km$^2$ in the fluctuating manner. Low and relatively low ecological risk regions were mainly in western and southern regions in the Yongjiang River Basin. High and relatively high ecological risk regions concentrated in northern and eastern regions of the basin, while the middle ecological risk region was a stripped zone that run from the east to the west and distributed in the middle of the basin. Facing with increasingly intensifying ecological risk in the Yongjiang River Basin, it is urgent to increase awareness of ecological environment protection, pay attentions to reasonable utilization and protection of Cultivated Land, Forest Land, Grassland, Water Areas, and Unused Land; make scientific planning of Construction Land; and promote coordinated economic development and ecological environment.

(4) Low, relatively low, middle, and relatively high ecological risk regions were mainly transferred to the next high grade of ecological risk regions. The area of ecological risk transfer from relatively high to high was the highest, reaching 813.07 km$^2$. Hotspot transfer regions were mainly in northern and eastern regions of the study area. There were 15 transfer directions among different ecological risk regions. Among them, areas of ecological risk transfer from middle to relatively high and from relatively high to high were large.

**Author Contributions:** Conceptualization, P.T. and J.L.; methodology, P.T.; software, P.T.; validation, J.L. and H.G.; formal analysis, P.T.; investigation, P.T.; resources, J.L.; data curation, P.T.; writing—original draft preparation, P.T.; writing—review and editing, J.L., H.G., R.P. and L.C.; visualization, P.T.; supervision, S.S., Z.S., X.F., L.W., and R.L..; project administration, J.L. and H.G.; funding acquisition, J.L. and H.G.

**Funding:** This research project was funded by the Zhejiang Natural Science Funds (LY17G030011), National Natural Science Funded project (71874091), NSFC-Zhejiang Joint Fund for the Integration of Industrialization and Informatization (U1609203), The major program of National Social Science Fund of China(16ZDA050), Zhejiang Foundation of Philosophy and Social Science (18NDJC095YB), Natural Science Foundation of Zhejiang Province (LQ15D020001) and the K.C. Wong Magna Fund of Ningbo University.

**Conflicts of Interest:** The authors declare no conflict of interest.

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
