# Peer review of "Research on Land Use Changes and Ecological Risk Assessment in Yongjiang River Basin in Zhejiang Province, China"

_sustainability, doi:10.3390/su11102817_

Round 1

Reviewer 1 Report

With considerations to impacts of land use changes on landscape pattern in the Yongjiang River Basin, in this paper some ecological risk assessment indexes were constructed and used to analyzed temporal and spatial variation characteristics of ecological risk at different time periods.

The paper deals important topics and it has a good structure, however it has some fundamental points of weekness:

a) the literature analisys should to be expanded more specifically to international literature (not only limited to studies about Chinese regions).

b) the indexes are "black boxes", because the authors are limited to indicating only bibliographical references. The indices should be better presented, in which cases they were used, characteristics and limitations, etc.

Good luck.

Author Response

With considerations to impacts of land use changes on landscape pattern in the Yongjiang River Basin, in this paper some ecological risk assessment indexes were constructed and used to analyzed temporal and spatial variation characteristics of ecological risk at different time periods.

The paper deals important topics and it has a good structure, however it has some fundamental points of weekness:

Point 1: the literature analisys should to be expanded more specifically to international literature (not only limited to studies about Chinese regions).

Response 1: Thanks for the suggestion.  In this revised version, a review on a large number of international literature was added in Introduction section (please see the changes on Page 1-4, Line 37-177).

Point 2: the indexes are "black boxes", because the authors are limited to indicating only bibliographical references. The indices should be better presented, in which cases they were used, characteristics and limitations, etc.

Response 2: Great comment. In the new version, we revised and added literature review and contents associated the “indices” in the introduction section, which included utilization, characteristics and limitations of the indices.  We also summarized the research history of ecological risk, the methods of ecological risk assessment and the purpose of selecting river basin as research object (see Page 1-4, Line 37-193).

Reviewer 2 Report

Review of the manuscript entitled Research on Land Use Changes and Ecological Risk Assessment in Yongjiang River Basin in Zhejiang Province, China

Dear Authors,
In general, research is of a high standard and is very interesting to me because it deals with similar issues. However, some additions are needed to improve the manuscript from a methodological point of view and to attract a wider audience.
First of all, the introduction section needs to be completed. There are no scientific questions at the end of this section that the research should answer or research hypotheses that the analysis should confirm or deny. This needs to be supplemented. It should be remembered, however, that answers to scientific questions or hypotheses should be included in the conclusions at the end of the manuscript.
In the section on methods and results of research, it would be interesting to supplement the analyses with the use of the Landscape Change Index (LCI), as the collected data make it possible. The calculation of the LCI is described in the following articles:

Krajewski, P.; Solecka, I.; Mrozik, K. Forest Landscape Change and Preliminary Study on Its Driving Forces in Ślęża Landscape Park (Southwestern Poland) in 1883–2013. Sustainability 2018, 10, 4526.

Krajewski P., Monitoring of Landscape Transformations within Landscape Parks in Poland in the 21st century. Sutainability 2019, (accepted for publication).       

Alternatively, it could be mentioned in the introduction or discussion section that such studies are being carried out and that it is also possible to use this indicator in an environmental risk assessment based on data on changes in the landscape pattern.
Figure 1 should be moved further to the left. Figure 4 shows maps for only 4 periods and Figure 6 for 7 periods. It is recommended that Figure 4 also shows changes in all periods. as can be seen, this is particularly relevant for the period 1995-2005 where land use has changed significantly. The absence of a map for the year 2000 is therefore necessary to complete the situation.

Author Response

In general, research is of a high standard and is very interesting to me because it deals with similar issues. However, some additions are needed to improve the manuscript from a methodological point of view and to attract a wider audience.

Point 1: First of all, the introduction section needs to be completed. There are no scientific questions at the end of this section that the research should answer or research hypotheses that the analysis should confirm or deny. This needs to be supplemented. It should be remembered, however, that answers to scientific questions or hypotheses should be included in the conclusions at the end of the manuscript.

Response 1: Thank you for the comment. In the Introduction section in the new version, we have expanded literature review, especially on international literature.  Usually, based on our research nature, we need to present several research objectives before ending the introduction section (thanks for reminding). Therefore, we have presented three research objectives clearly in the last paragraph in this section (thanks).  To address the research objectives, from this study derived results, in the Conclusion section, there are several conclusions summarized there to directly respond the three research objectives.

Point 2: In the section on methods and results of research, it would be interesting to supplement the analyses with the use of the Landscape Change Index (LCI), as the collected data make it possible. The calculation of the LCI is described in the following articles:

Krajewski, P.; Solecka, I.; Mrozik, K. Forest Landscape Change and Preliminary Study on Its Driving Forces in Ślęża Landscape Park (Southwestern Poland) in 1883–2013. Sustainability 2018, 10, 4526.

Krajewski P., Monitoring of Landscape Transformations within Landscape Parks in Poland in the 21st century. Sutainability 2019, (accepted for publication).      

Alternatively, it could be mentioned in the introduction or discussion section that such studies are being carried out and that it is also possible to use this indicator in an environmental risk assessment based on data on changes in the landscape pattern.

Response 2: Thank you for the comment. In the introduction, we added a review of Krajewski's research results, and analyzed the land use change in the Yongjiang River basin with landscape change index(LCI), which was also cited. (see Page4, Line 222-239; Page8, Line 309-330).

Point 3: Figure 1 should be moved further to the left. Figure 4 shows maps for only 4 periods and Figures 6 for 7 periods. It is recommended that Figure 4 also shows changes in all periods. as can be seen, this is particularly relevant for the period 1995-2005 where land use has changed significantly. The absence of a map for the year 2000 is therefore necessary to complete the situation.

Response 3: Thank you for the suggestion. In the revised version, we redid the Figure 1, so it was moved further to the left. We have made up missing the land use intensity map, so all periods maps there now.  Thanks.  (see Page5, Line 209; Page12, Line 395).

Round 2

Reviewer 1 Report

The authors have addressed sufficiently my comments. It's fine for me.

Good luck.

Reviewer 2 Report

Dear Authors,

The manuscript has been greatly improved and now has a more scientific soundness. However, one more element needs to be corrected. This concerns the supplementing of data on the use of the landscape change index. It is recommended that the manuscript be supplemented in the introduction section (L127-130) with information that the landscape change index is also used as a way of monitoring landscape changes and therefore can be used for landscape ecological risk assessment. At this point the following article should be referenced:
Krajewski, P. Monitoring of Landscape Transformations within Landscape Parks in Poland in the 21st Century. Sustainability 2019, 11, 2410.

Author Response

Point 1: The manuscript has been greatly improved and now has a more scientific soundness. However, one more element needs to be corrected. This concerns the supplementing of data on the use of the landscape change index. It is recommended that the manuscript be supplemented in the introduction section (L127-130) with information that the landscape change index is also used as a way of monitoring landscape changes and therefore can be used for landscape ecological risk assessment. At this point the following article should be referenced:

Krajewski, P. Monitoring of Landscape Transformations within Landscape Parks in Poland in the 21st Century. Sustainability 2019, 11, 2410.

Response 1: Thank you for the comment. As the experts said, this study also shows that landscape change index can be used as a method to monitor landscape changes, and therefore can be used for landscape ecological risk assessment, so I added this sentence in the paper and quoted this paper. (please see the changes on Page 3, Line 127-132)